# Selective sulfur dioxide adsorption on crystal defect sites on an isoreticular metal organic framework series

L. Marleny Rodríguez-Albelo[1], Elena López-Maya[1], Said Hamad[2], A. Rabdel Ruiz-Salvador[2], Sofia Calero[2] & Jorge A.R. Navarro[1]

The widespread emissions of toxic gases from fossil fuel combustion represent major welfare risks. Here we report the improvement of the selective sulfur dioxide capture from flue gas emissions of isoreticular nickel pyrazolate metal organic frameworks through the sequential introduction of missing-linker defects and extra-framework barium cations. The results and feasibility of the defect pore engineering carried out are quantified through a combination of dynamic adsorption experiments, X-ray diffraction, electron microscopy and density functional theory calculations. The increased sulfur dioxide adsorption capacities and energies as well as the sulfur dioxide/carbon dioxide partition coefficients values of defective materials compared to original non-defective ones are related to the missing linkers enhanced pore accessibility and to the specificity of sulfur dioxide interactions with crystal defect sites. The selective sulfur dioxide adsorption on defects indicates the potential of fine-tuning the functional properties of metal organic frameworks through the deliberate creation of defects.

[1] Departamento de Química Inorgánica, Universidad de Granada, Av. Fuentenueva S/N, 18071 Granada, Spain. [2] Department of Physical, Chemical and Natural Systems, Universidad Pablo de Olavide, Ctra. de Utrera, Km. 1, 41013 Sevilla, Spain. Correspondence and requests for materials should be addressed to L.M.R.-A. (email: mralbelo@ugr.es) or to J.A.R.N. (email: jarn@ugr.es).

Developing efficient technologies for capturing $CO_2$, $NO_x$ and $SO_x$ gases from static and mobile sources represents a major challenge for creating a cleaner and healthier environment[1,2]. The removal of these acidic gases from power plant flue gas emissions leads to the extensive use of alkaline aqueous solutions. Solid adsorbent materials are sought after to prevent problems associated with leaks of resultant toxic and corrosive solutions. Though zeolites have been used for this purpose, their relatively small pores limit their suitability for $NO_x$ and $SO_x$ abatement[3,4]. In this respect, the high structural and functional flexibility of metal organic frameworks (MOFs) makes them attractive for reducing greenhouse and toxic gas exhausts[5–9]. For environmental applications, stable materials that can maintain their functional properties under harsh conditions are especially desired. In MOF design, the strength of the metal–ligand bond plays a fundamental role in stability. The appropriate combination of metal-donor atoms to enhance hard–hard (that is, Zr(IV)-O) or soft–soft (that is, Ni(II)-N) acid–base interactions should result in more resilient porous materials[10–13]. Besides stability, improving material functionality is key to obtaining useful MOFs for environmental applications. In this regard, one emerging strategy to improve the functional properties of MOFs[14–16] is the deliberate introduction of defects. For example, the defects in $[Zr_6O_4(OH)_4(bdc)_6]$ UiO-66 frameworks allow the incorporation of active acid and basic sites ultimately leading to enhanced capture[17,18] and catalytic degradation of toxic gases[19].

We have recently shown that the postsynthetic treatment of the $[Ni_8(OH)_4(H_2O)_2(BDP\_X)_6]$ $(H_2BDP\_X = 1,4$-bis(pyrazol-4-yl) benzene-4-X with X = H (**1**), OH (**2**), $NH_2$ (**3**)) systems with ethanolic solutions of potassium hydroxide leads to the formation of defective $K[Ni_8(OH)_3(EtO)_3(BDP\_X)_{5.5}]$ (**1@KOH**, **3@KOH**) and $K_3[Ni_8(OH)_3(EtO)(BDP\_O)_5]$ (**2@KOH**). The latter materials exhibit improved carbon dioxide capture[20] and ion conductive[21] properties related to missing-linker enhanced accessibility of the porous network, higher basicity of the metal clusters and the presence of extra-framework cations.

In the present work, we target the selective adsorption of sulfur dioxide, from power plant exhaust gases. With this aim, we introduce extra-framework $Ba^{2+}$ ions into the porous structure of the defective nickel pyrazolate systems, by means of an ion exchange process. The strategy is completed with the assistance of ligand functionalization (Fig. 1). To rationalize the performance of the materials on sulfur dioxide capture, we carry out a joint experimental and theoretical study.

## Results

**Introduction of extra-framework barium ions on defective MOFs.** The presence of extra-framework cations in the $K[Ni_8(OH)_3(EtO)_3(BDP\_X)_{5.5}]$ (**1@KOH**; **3@KOH**) and $K_3[Ni_8(OH)_3(EtO)(BDP\_O)_5]$ (**2@KOH**) systems opens the way to ion exchange processes. Exposing **1@KOH–3@KOH** materials to aqueous $Ba(NO_3)_2$ solutions incorporates barium into the solids, yielding the defective ion exchanged $Ba_{0.5}[Ni_8(OH)_3(EtO)_3(BDP\_X)_{5.5}]$ (**1@Ba(OH)$_2$**, X = H; **3@Ba(OH)$_2$**, X = $NH_2$), and $Ba_{1.5}[Ni_8(OH)_3(EtO)(BDP\_O)_5]$ (**2@Ba(OH)$_2$**) systems (Figs 1 and 2). Inductively coupled plasma (ICP), high resolution transmission electron microscopy (HRTEM) and energy-dispersive X-ray (EDX) analyses confirmed the complete exchange of potassium by barium (Fig. 2d, Supplementary Figs 1–6). Moreover, compared with defective **1–3@KOH** MOFs, the porosity of the barium exchanged materials is maintained with a slight decrease in porous network accessibility (Fig. 2c).

Characterization by X-ray powder diffraction (XRPD) shows that the nickel pyrazolate frameworks retain their crystallinity

despite of the concomitant changes from a neutral to an anionic framework that result from the missing linker defects and the entrance of potassium extra-framework cations and its subsequent exchange by barium cations (Fig. 2b). Le Bail profile fittings of XRPD patterns of defective MOFs (Supplementary Fig. 7, Supplementary Tables 1,2 and Supplementary Methods) confirm that cell volume variations upon postsynthetic modifications (PSMs) of **1–3** systems are below 1% and that the overall cubic symmetry is preserved. Two partial conclusions emerge from the analysis, first the high stability of the nickel pyrazolate-based MOFs is demonstrated and second it is shown that the missing linkers are likely to be randomly distributed along the 3D framework. Moreover, XRPD and HRTEM-EDX experiments show that $Ba(OH)_2$ co-crystallizes with the defective MOFs after barium ion exchange process. Aggregates with an approximate 1:1 molar ratio of $Ba_{0.5}[Ni_8(OH)_3(EtO)_3(BDP\_X)_{5.5}]$ and $Ba(OH)_2$ nano/microcrystals form. The formation of aggregates arises from the high hydroxide content of the metal clusters which increases the basicity of the external surface of the MOF particles. This basicity facilitates the crystal growth of $Ba(OH)_2$ on MOF grain boundaries (Supplementary Figs 1,5 and 6).

To gain some insight into the structural effects caused by both the introduction of defects and the subsequent ion exchange process, we have carried out periodic density functional theory (DFT) calculations[22]. Thus far, molecular modelling of defective MOF materials has received minor attention, as a consequence of the expensive DFT calculations required for accurate studies[23–26]. Therefore, we have carefully selected a range of defect environments that would be wide enough to yield reliable models of the defective MOF frameworks, which could help to interpret the experimental data (Supplementary Note 1 and Supplementary Methods), while at the same time being computationally affordable. The structure of pristine system **1** was used as a starting point for the geometrical optimization of the defective frameworks of **1@KOH** and **1@Ba(OH)$_2$** materials. The symmetry constraints imposed by the space group ($Fm$-$3m$) of the crystallographic reported structure[12] were removed, to model the local crystal defect sites in **1@KOH** and **1@Ba(OH)$_2$** systems. The calculations show that after PSMs some distortions are noticed, namely, $\pm 0.25$ Å deviations in the intracluster Ni–Ni distances (Fig. 2a, Supplementary Table 6). This is not surprising, as a previous spectroscopy study showed that lowering of the local symmetry of the octanuclear nickel hydroxide cluster takes place upon dehydration[27]. Moreover, DFT results place the extra-framework cations close to crystal defect sites. The cations establish short contacts with terminal hydroxide group welded by the cluster on the missing linker site (Fig. 2a). In sum, although the introduction of missing linker defects and extra-framework cations are responsible for the observed distortions, the average 3D structure of the MOF framework is preserved.

**Sulfur dioxide adsorption.** Dynamic adsorption experiments reveal the impact of the deliberate introduction of defects and subsequent $K^+$ to $Ba^{2+}$ ion exchange process on the $SO_2$ capture. We used breakthrough curve measurements of simulated flue gas to determine the $SO_2$ adsorption capacity and selectivity of the materials under study. In addition, we carried out variable temperature pulse gas chromatography to determine the thermodynamic parameters governing the physisorption of $SO_2$.

Measurement of breakthrough curves, using a $20\,ml\,min^{-1}$ flow of $N_2/SO_2$ (97.5:2.5) gas mixture at 303 K, evaluated the capability of our materials to selectively capture $SO_2$ (Fig. 3a–d). All materials were activated at 423 K, in helium flow ($20\,ml\,min^{-1}$) for 24 h, before the first adsorption cycle, and for 2 h between the successive breakthrough cycles. The $SO_2$

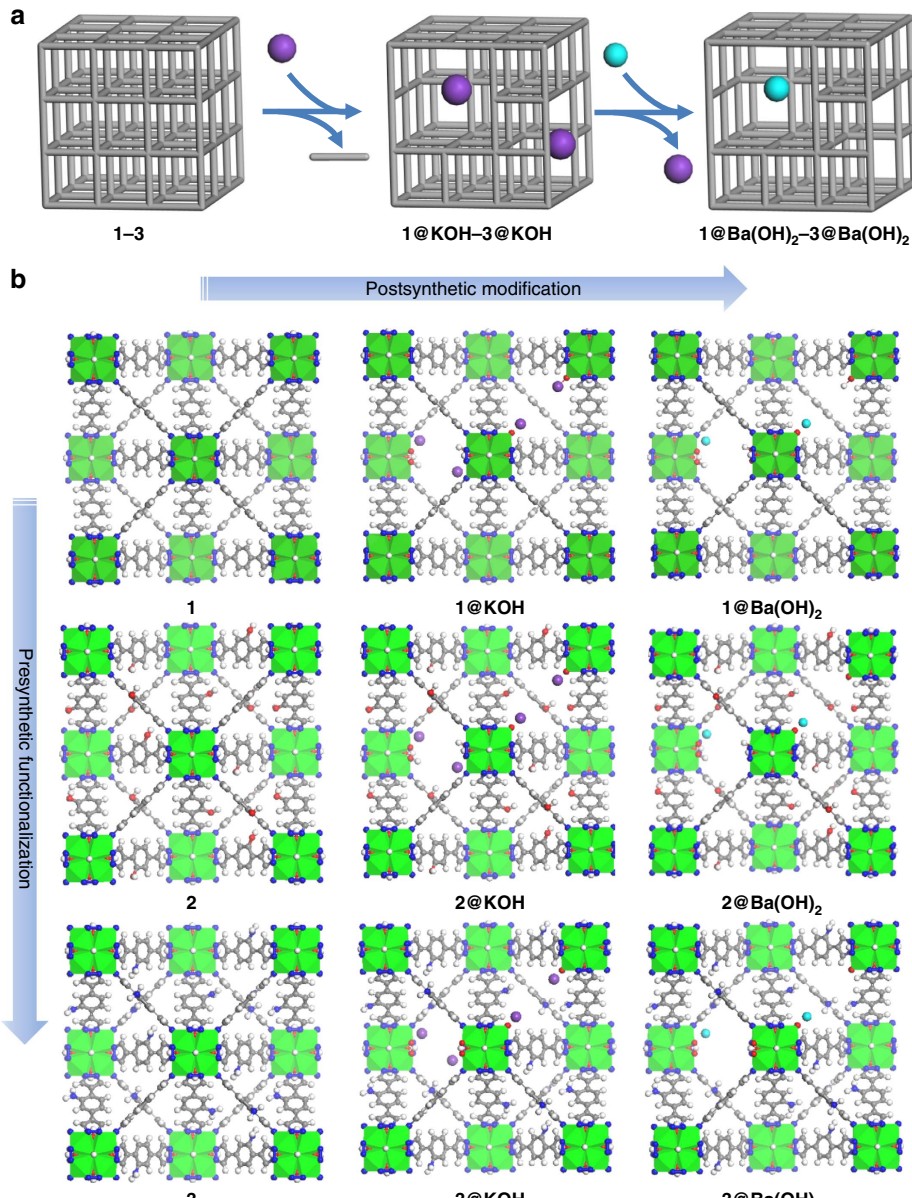

**Figure 1 | Defect pore engineering in an isoreticular metal organic framework series.** (**a**) Schematic representation of the successive PSMs, from pristine nickel pyrazolate $[Ni_8(OH)_4(H_2O)_2(BDP\_X)_6]$ ($H_2BDP\_X = 1,4$-bis(pyrazol-4-yl)benzene-4-X with $X = H$ (**1**), OH (**2**), $NH_2$ (**3**)) frameworks to yield the missing linker defective $K[Ni_8(OH)_3(EtO)_3(BDP\_X)_{5.5}]$ (**1@KOH**, **3@KOH**) and $K_3[Ni_8(OH)_3(EtO)(BDP\_O)_5]$ (**2@KOH**) and subsequently, the ion exchanged $Ba_{0.5}[Ni_8(OH)_3(EtO)_3(BDP\_X)_{5.5}]$ (**1@Ba(OH)₂**, $X = H$; **3@Ba(OH)₂**, $X = NH_2$), and $Ba_{1.5}[Ni_8(OH)_3(EtO)(BDP\_O)_5]$ (**2@Ba(OH)₂**) materials. Organic linker (grey bar), potassium (purple), barium (cyan). (**b**) Overview of the structures of the studied materials used for the selective capture of $SO_2$. The presynthetic functionalization of the organic spacers with –OH and –$NH_2$ polar tags are shown top to bottom. The successive PSMs are shown left to right. Ni (green octahedra); K (purple); Ba (cyan); C (grey); N (blue); O (red); H (white).

adsorption capacities at 303 K and at a low $SO_2$ partial pressure of 0.025 bar exhibited by our materials are considerably high, ranging from 2 mmol g$^{-1}$ for **1** to 5.6 mmol g$^{-1}$ for **3@Ba(OH)₂** (Fig. 3e and Supplementary Table 3). These results are indicative that both the presynthetic introduction of amino and hydroxyl functional groups on the organic linkers and the PSMs, synergistically increase $SO_2$ capture capacity. Moreover, some degree of irreversible, chemical sorption of $SO_2$, is observed in all tested systems, which is manifested by a certain reduction in the adsorption capacity between the first and successive breakthrough cycles. This observation is particularly evident in the defective solids), although, 63–74% of the $SO_2$ molecules are physisorbed, in a reversible manner (Fig. 3 and Supplementary Table 3). In this

regard, it is worth noting that the 3.7 mmol g$^{-1}$ reversible adsorption capacity achieved by **3@Ba(OH)₂** at 303 K and 25 mbar is in line with the behaviour of the best MOF materials for $SO_2$ adsorption, namely, 1 mmol g$^{-1}$ achieved by NOTT-202a (ref. 7) at 283 K and 25 mbar, although, slightly below the 4.13 mmol g$^{-1}$ and 5.40 mmol g$^{-1}$ values recently reported for MTM-300 (ref. 28) at 298 K and 25 mbar and NOTT-300 (ref. 6) at 303 K and 25 mbar, respectively. To further prove the extent at which reversible $SO_2$ physisorption takes place on **1@Ba(OH)₂** we have performed 10 successive adsorption/desorption cycles of $SO_2$ adsorption (See Fig. 3f). The results show that after all available sites for $SO_2$ chemisorption are occupied, in the first cycle, reversible adsorption takes place in a steady way proving

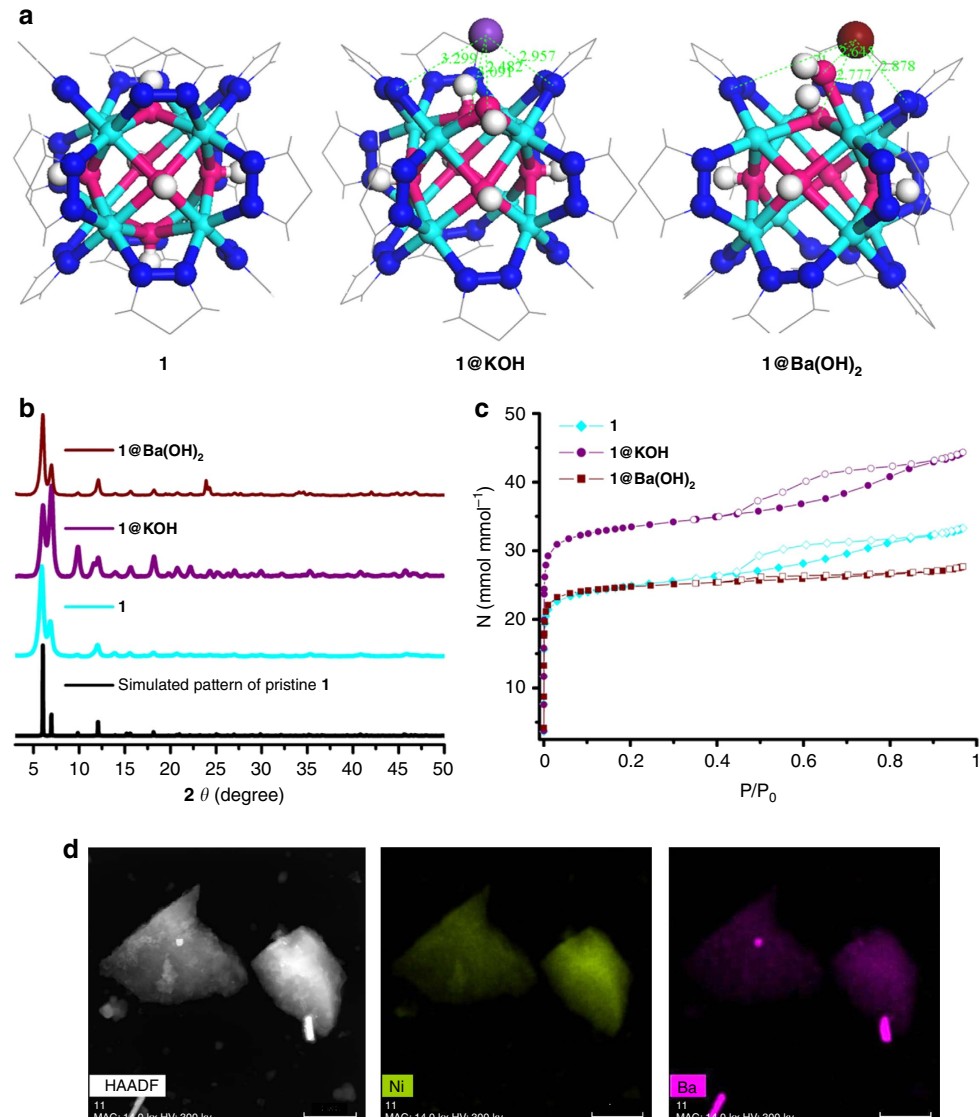

**Figure 2 | Structural characterization and computational modelling.** (**a**) Detail of the crystal structure of **1** around the metal cluster[12] and DFT optimized structures of the crystal defects in **1@KOH**, and ion exchanged **1@Ba(OH)₂**. Ni (cyan); K (purple); Ba (wine); N (blue); O (magenta); H (white); C (grey lines). (**b**) XRPD patterns for **1** and its **1@KOH** and **1@Ba(OH)₂** derivative materials. (**c**) Impact of PSMs on the accessibility of the pore structure to nitrogen probe molecule at 77 K on the **1**, **1@KOH**, **1@Ba(OH)₂** series. (**d**) HRTEM images of sample **1@Ba(OH)₂** in the high angle annular dark-field (HAADF) and EDX mapping for Ni and Ba elements. The scale bar in (**d**) is referred to 1 μm.

materials stability upon continuous $SO_2$ exposure. Besides, an adsorption isotherm of $SO_2$ was measured for **1@Ba(OH)₂** at 303 K (Supplementary Fig. 8) being indicative of adsorption reversibility, since this measurement implied successive reactivation of the material at each measured point. Moreover, the XRPD pattern of **1@Ba(OH)₂** material before, during and after $SO_2$ adsorption is maintained which is in agreement with materials stability towards $SO_2$ (Supplementary Fig. 12). In addition, the infrared spectra of **1@Ba(OH)₂** before, during and after the adsorption of $SO_2$ were measured. Noteworthy, the infrared spectra of $SO_2$ loaded **1@Ba(OH)₂** material shows two sets of vibrational bands ($\gamma_{sym}$, $\gamma_{assym}$) of adsorbed $SO_2$, the first set of strong intensity is located at 1,140, 1,328 cm$^{-1}$ and the second set of weak intensity is located at 1,148, 1,385 cm$^{-1}$ (Supplementary Fig. 10). While the first set of bands disappears upon reactivation of the material, being indicative of physisorbed $SO_2$ (ref. 29), the second set of bands is maintained, being indicative of chemisorbed $SO_2$ (see below). These results suggest

that the reversible physisorption process dominate the adsorption behaviour of these materials series.

The affinity of the studied systems for $SO_2$ adsorption can be justified on the basis of the increased basicity of the nickel hydroxide clusters after the introduction of defects. Lewis acid–base interactions with $SO_2$ molecules[1,30,31] lead to the formation of $HSO_3^-$ and $SO_3^{2-}$ species according to equation 1:

$$\left[Ni_8(OH)_6(OH)_2\right]^{10+} + SO_2 \rightarrow \left[Ni_8(OH)_5(HSO_3)\right]^{10+}$$
$$\rightarrow \left[Ni_8(OH)_4(SO_3)(H_2O)\right]^{10+} \quad (1)$$

Moreover, the presence of extra-framework cations located close to defects sites is likely to promote the formation of more stable species of the $MSO_3$ (M = Ba, 2 K) type according to equation 2:

$$M\left[Ni_8(OH)_4(SO_3)(H_2O)(BDP\_X)_{5.5}\right]^+$$
$$\rightarrow (MSO_3) \cdot \left[Ni_8(OH)_4(H_2O)(BDP\_X)_{5.5}\right]^+ \quad (2)$$

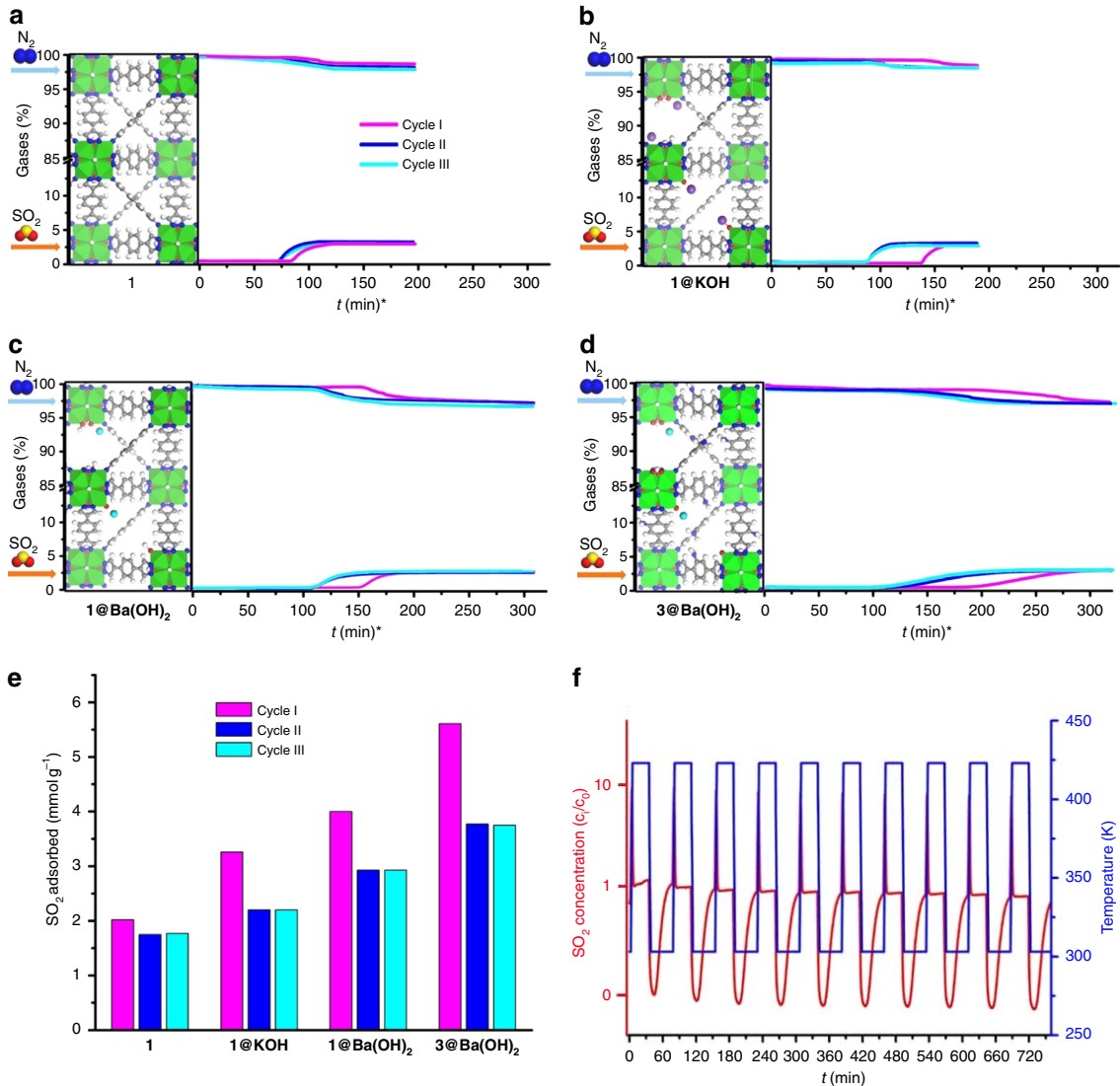

**Figure 3 | SO₂ capture from flue gas.** Breakthrough curves with 20 ml min⁻¹ flow of N₂/SO₂ (97.5:2.5) gas mixture for **1** (**a**), **1@KOH** (**b**), **1@Ba(OH)₂** (**c**) and **3@Ba(OH)₂** (**d**) materials at 303 K showing the effect of both presynthetic and PSMs on the improved capture of SO₂ (*time relative to 1 g of MOF). (**e**) Bar graph summary of SO₂ adsorption capacities of the studied MOFs over three successive cycles of breakthrough curves, and (**f**) activation/adsorption process over multiple cycles on **1@Ba(OH)₂** under a constant flow of 20 ml min⁻¹ of N₂/SO₂ (97.5:2.5) gas mixture.

These proposed chemisorption mechanisms are supported by the DFT simulations of the interaction of SO₂ with the solids (see below). From an experimental point of view, in addition to the above breakthrough curves, the presence of chemical adsorption is confirmed by a number of techniques (Methods and Supplementary Figs 10–13). Elemental analyses show the occurrence of the SO₂ chemisorption process in the **MOF@Ba(OH)₂** materials showing 0.2–0.5:1 S:metal cluster content ratio (Supplementary Methods). In addition, a detailed inspection of FTIR spectra of **MOF@Ba(OH)₂** after SO₂ adsorption reveals two weak absorption bands at 1,148 and 1,385 cm⁻¹, which correspond to symmetric and asymmetric stretching bands ($\gamma_{sym}$, $\gamma_{assym}$) of $SO_3^{2-}$ groups (Supplementary Fig. 10). FTIR analysis of evolved gases during thermogravimetric analysis experiments under an inert atmosphere indicates SO₂ evolution in the 511–594 K temperature range, ca. 80 K before materials pyrolysis (Supplementary Fig. 11). To further confirm the presence of $SO_3^{2-}$ species in the **MOF@Ba(OH)₂** materials, we exposed them to a humid air flow, which oxidized the $SO_3^{2-}$ groups to $SO_4^{2-}$ groups, as evidenced by a new FTIR vibration band at 1,100 cm⁻¹

(Supplementary Fig. 10). XRPD analysis (Supplementary Fig. 12) and nitrogen adsorption (Supplementary Fig. 13 and Supplementary Table 5) experiments of the materials after breakthrough measurements and subsequent activation at 423 K indicate that the MOFs retain their characteristic porosity and crystallinity. Noteworthy, the study of the materials after breakthrough tests also points to a clear difference between the reactivity of extra-framework barium cations in the MOF structure and the barium cations in co-crystallized barium hydroxide particles. HRTEM-EDX mapping analyses of the samples after breakthrough tests shows that the S content is only found related to Ba/Ni homogeneous distribution in MOF particles (Fig. 5e). By contrast, the mapping analysis of Ba(OH)₂ nano/microparticles do not show any S content. Moreover, the XRPD pattern of this sample did not show any changes in the intensity/position of the Ba(OH)₂ reflections, before, during and after SO₂ adsorption (Supplementary Fig. 12), which further evidences that SO₂ capture occurs solely in the MOF particles.

As the performance of the materials in separating complex flue gas mixtures is relevant from the applications point of view, the

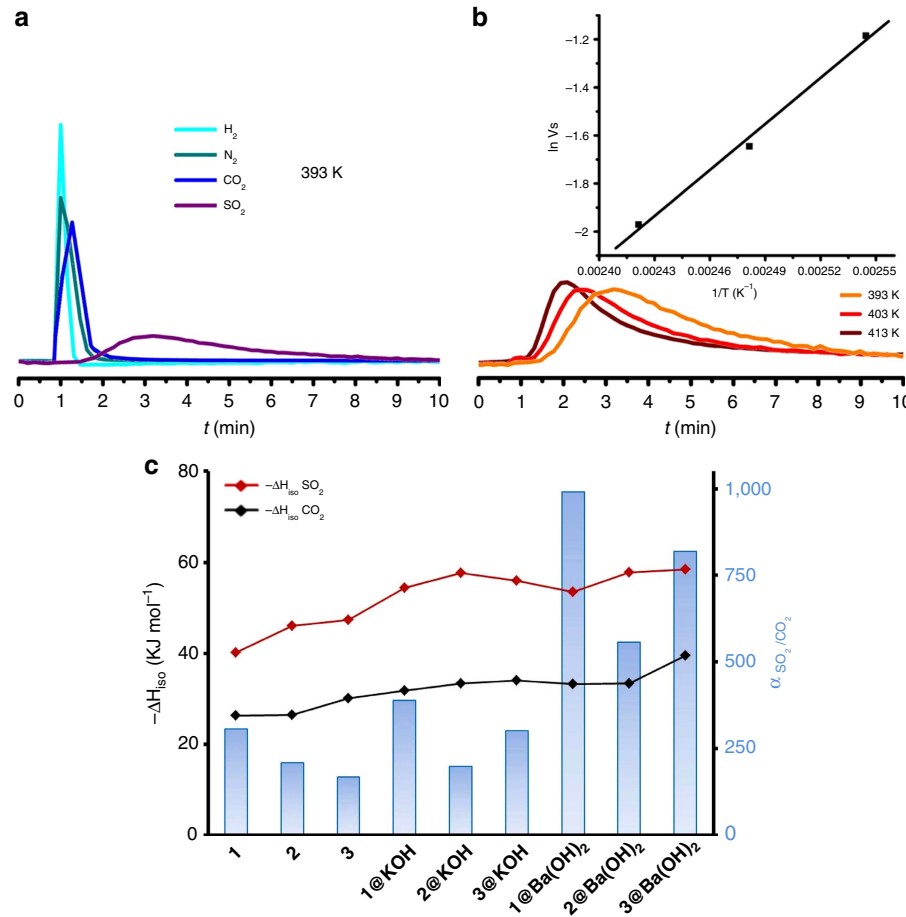

**Figure 4 | Thermodynamic characterization of SO₂ and CO₂ adsorption processes.** Variable temperature pulse gas chromatography results as: (**a**) Chromatogram of equimolecular mixture of $H_2$, $N_2$, $CO_2$ and $SO_2$ passed through a column packed with **1@Ba(OH)₂**, using a He flow of 20 ml min$^{-1}$ at 393 K; (**b**) Thermal dependence of the retention time of $SO_2$ for **1@Ba(OH)₂**. The inset in (**b**) shows the fitting of the variation of the retention volume Vs (in cm³ g$^{-1}$) as a function of the adsorption temperature (393–413 K). (**c**) Comparison of isosteric enthalpy ($\Delta H_{iso}$) values for $SO_2$ and $CO_2$ adsorption and $\alpha_{SO_2/CO_2}$ partition coefficients for the essayed **1–3@Ba(OH)₂** materials. It should be noted that all the measurements were done on the materials after $SO_2$ chemisorption to ensure thermodynamic equilibrium.

**Table 1 | Specific SA and thermodynamic parameters.**

| MOF | SA (m² g⁻¹) | − ΔHiso (kJ mol⁻¹) | − ΔS (J K⁻¹ mol⁻¹) | − ΔG (kJ mol⁻¹*) | $\alpha_{SO_2/CO_2}$* |
|---|---|---|---|---|---|
| **1** | 1,735 | 40.1 | 107.1 | 8.2 | 310 |
| **2** | 1,385 | 46.0 | 123.3 | 9.2 | 210 |
| **3** | 1,170 | 47.4 | 124.6 | 10.3 | 170 |
| **1@KOH** | 2,055 | 54.3 | 142.7 | 11.8 | 390 |
| **2@KOH** | 1,830 | 57.7 | 150.1 | 12.9 | 200 |
| **3@KOH** | 1,665 | 56.0 | 146.7 | 12.3 | 300 |
| **1@Ba(OH)₂** | 1,250 | 53.5 | 137.5 | 12.5 | 990 |
| **2@Ba(OH)₂** | 850 | 57.8 | 147.8 | 13.7 | 560 |
| **3@Ba(OH)₂** | 1,190 | 58.4 | 150.4 | 13.6 | 820 |

Summary of specific SA and thermodynamic parameters for SO₂ adsorption and on **1–3**, **1@KOH–3@KOH** and **1@Ba(OH)₂–3@Ba(OH)₂** materials series. Thermodynamic parameters of enthalpy, entropy and Gibbs free energy for SO₂ adsorption and $\alpha_{SO_2/CO_2}$ partition coefficients have been obtained from van't Hoff plotting of the chromatographic results on the materials after SO₂ chemisorption to ensure the thermodynamic equilibrium (for CO₂ see Supplementary Table 3). SA, surface area.
*Calculated values at 298 K.

**1@Ba(OH)₂** selectivity for $SO_2$ capture was evaluated by using breakthrough experiments (Supplementary Fig. 9). Typical gas mixtures compositions were used, namely, 20 ml min$^{-1}$ flow of $N_2/CO_2/SO_2$ (83.5:14:2.5) and $N_2/H_2O/SO_2$ (94.1:3.4:2.5). The presence of $CO_2$ and/or moisture does not significantly affect $SO_2$ adsorption in **1@Ba(OH)₂**. Before discussing the quantitative details of the adsorption processes in these mixtures, it is important to highlight that only physisorption is observed for

both $CO_2$ and $H_2O$ molecules in **1@Ba(OH)₂**, in contrast to $SO_2$ which shows both chemical and physical adsorption (Supplementary Fig. 9). The presence of 15% of $CO_2$ versus 2.5% of $SO_2$ in the gas mixture is responsible for a diminution ($\sim 30\%$) in the adsorption capacity of $SO_2$ compared with the previous experiments without $CO_2$ (Fig. 3). This result is in agreement with the high $\alpha_{SO_2/CO_2}$ partition coefficient values of PSMs MOFs (see below). Similarly, **1@Ba(OH)₂** is still able to

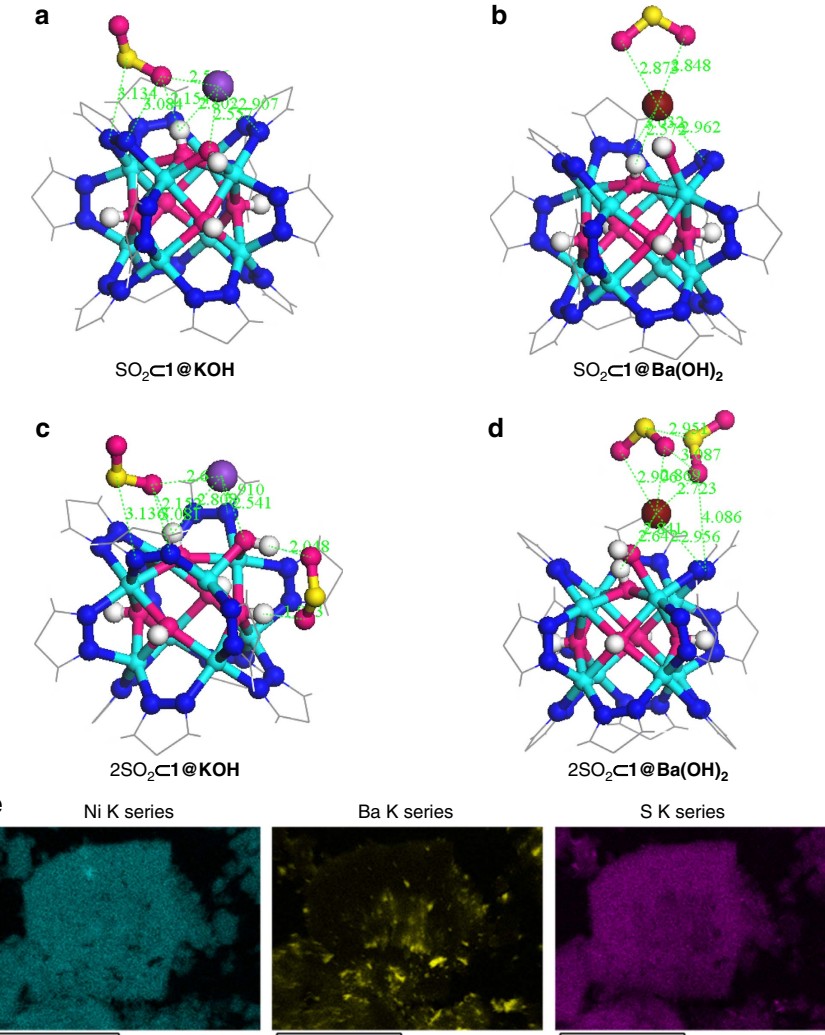

**Figure 5 | Sulfur dioxide interaction with crystal defect sites.** DFT structure minimization of molecular configuration of one (**a,b**) and two (**c,d**) adsorbed SO₂ molecules on **1@KOH** (left) and **1@Ba(OH)₂** (right) materials. For sake of clarity, only the region around the metal cluster is shown. Ni (cyan); K (purple); Ba (wine); C (grey); N (blue); O (magenta); H (white); S (yellow). (**e**) HRTEM-EDX mapping images for **1@Ba(OH)₂** after SO₂ adsorption showing (from left to right) the individual distributions of Ni, Ba and S elements. The scale bar in (**e**) is referred to 25 μm.

adsorb significant amounts of $SO_2$ under 80% relative humidity (0.034 bar), as 60% of the adsorption capacity is retained as compared with dry conditions. Shielding of the extra-framework $Ba^{2+}$ cations and $SO_2$ interactions caused by competitive hydration of the $Ba^{2+}$ cations can explain the observed decrease (see DFT discussion below). Overall, it is observed that the amounts of $SO_2$ captured in gas mixtures of practical interest are high.

Our experiments have shown that reversible $SO_2$ adsorption working capacities are in the 1.8–3.7 mmol g$^{-1}$ range. Since these values represent the larger fraction of captured $SO_2$ in our materials, it is important to have a better understanding of the role of pre- and PSMs on the strength of $SO_2$ interactions with the MOFs pore surface. To do that, we performed gas chromatography experiments using activated MOFs as chromatographic beds. The experiments were conducted in the 303–423 K temperature range, and a pulse of an equimolecular mixture of $H_2$, $N_2$, $CO_2$ and $SO_2$ gases was employed. These experiments were carried out on the materials after the irreversible $SO_2$ chemisorption process (breakthrough curve experiments), to ensure thermodynamic equilibrium (reversibility) for $CO_2$ and $SO_2$ adsorption. The chromatograms were processed using van't Hoff analysis to calculate the thermodynamic parameters for $CO_2$ and $SO_2$ adsorption processes (Fig. 4, Table 1, Supplementary

Table 4 and Supplementary Methods). The observed retention times follow the trend $H_2 \sim N_2 < CO_2 \ll SO_2$, which suggests a strong interaction of the $SO_2$ molecules with the pore structure, followed by significant interaction with $CO_2$ and negligible interactions with $N_2$ and $H_2$ molecules. The isosteric heat of adsorption and Gibbs free energy values for $SO_2$ typically increase by circa 10–14 kJ mol$^{-1}$ and 2–4 kJ mol$^{-1}$, respectively, in the defective solids, suggesting the presence of strong interactions between adsorbed $SO_2$ molecules and the PSM porous frameworks along the **1–3**, **1@KOH–3@KOH**, **1@Ba(OH)₂–3@Ba(OH)₂** series (Fig. 4, Table 1). A similar behaviour is observed for $CO_2$, for which the enthalpy values increase by 7–19 kJ mol$^{-1}$ along the **1–3@Ba(OH)₂** series (Supplementary Table 4). The selective interaction of $SO_2$ with the PSM frameworks is also illustrated by a four-fold enhancement of $\alpha_{SO_2/CO_2}$ partition coefficients on passing from **1** to **3@Ba(OH)₂** systems (Fig. 4). Indeed, these high $\alpha_{SO_2/CO_2}$ partition coefficient values of the PSM materials can be related to the increasing differences in the isosteric heat of adsorption values between $SO_2$ and $CO_2$ $|\Delta H_{SO_2} - \Delta H_{CO_2}|$, 13.8 kJ mol$^{-1}$ for **1** versus 18.8 kJ mol$^{-1}$ for **3@Ba(OH)₂** (Fig. 4). This positive effect correlates with the incorporation of extra-framework $Ba^{2+}$ ions, and the enhanced basicity related to the presence of additional hydroxide

ligands substituting the linker vacancies in the metal clusters. Consequently, defects and ion exchange processes enhance $SO_2$ interactions with MOF pores. The introduction of –$NH_2$ and –OH functionalities in the linkers also has a non-negligible impact on $SO_2$ interactions, as manifested by increased enthalpy values for the **2@Ba(OH)$_2$** and **3@Ba(OH)$_2$** systems compared to **1@Ba(OH)$_2$** (that is, |$\Delta H_{SO_2}$(**3@Ba(OH)$_2$**)-$\Delta H_{SO_2}$(**1@Ba(OH)$_2$**)| = 5 kJ mol$^{-1}$). These observations agree with the favourable -$NH_2 \cdots SO_2$ and $OH \cdots SO_2$ interactions found in NOTT-202a system[7]. Although the introduction of polar functionalization results in high $\alpha_{SO_2/CO_2}$ partition coefficients, the selectivity for $SO_2$ over $CO_2$ is slightly diminished, particularly for **2@Ba(OH)$_2$**, because of the strong interactions of $CO_2$ molecules with **2@KOH** system[20], as manifested by lower $\alpha_{SO_2/CO_2}$ for **2@Ba(OH)$_2$** and **3@Ba(OH)$_2$** materials compared with **1@Ba(OH)$_2$** (Fig. 4, Table 1, Supplementary Table 4). A related effect has previously been observed in the $C_2H_2/CO_2$ separation from $C_2H_2/CO_2/N_2$ mixtures by [M(BDP_X)] systems (M = Ni, Zn; X = $NO_2$, $NH_2$, OH, $SO_3H$)[32]. Still, the $SO_2$ adsorption enthalpy and $\alpha_{SO_2/CO_2}$ values for **1@Ba(OH)$_2$**–**3@Ba(OH)$_2$** materials are, respectively, 20 kJ mol$^{-1}$ and 1 order of magnitude higher than those found for MTM/NOTT-300 (refs 6,28) materials exhibiting record $SO_2$ adsorption capacities. Thus, while the irreversible $SO_2$ chemisorption on the **1@Ba(OH)$_2$**–**3@Ba(OH)$_2$** systems give rise to slightly lower working capacities than MTM/NOTT-300 (ref. 28) materials (see above), the increased $SO_2$ adsorption selectivity of the PSM nickel pyrazolate series, clearly benefits their performance in gas separation applications.

**Interaction of $SO_2$ with crystal defects by DFT calculations.** To better understand the adsorption mechanisms of $SO_2$ on the defective structures, we explored the preferential adsorption configurations of one, two and three $SO_2$ molecules in the pore structures of **1**, **1@KOH** and **1@Ba(OH)$_2$** materials by means of DFT calculations[22] (Fig. 5 and Supplementary Fig. 14). The DFT results show that the preferential $SO_2$ adsorption sites in **1@Ba(OH)$_2$** are the crystal defects. Indeed, the first $SO_2$ molecule adsorbed exhibits a bidentate coordination of its oxygen donor atoms with the extra-framework $Ba^{2+}$ cations, which are located at the pyrazolate vacancy sites with Ba–O distances of ca. 2.9 Å (Fig. 5a). When a second $SO_2$ molecule is adsorbed, the first molecule keeps its bidentate coordination; however, the second molecule interacts in a monodentate mode through one oxygen donor atom giving rise to a shorter 2.7 Å Ba $\cdots$ O contact (Fig. 5c). The binding energy for the first molecule ( $-107.4$ kJ mol$^{-1}$ ) is therefore higher than for the second one ( $-78.2$ kJ mol$^{-1}$, Supplementary Table 7). These high energy values are indicative of strong and selective interactions, namely, chemisorption[33]. Since the first two molecules have such high adsorption energies, we studied the adsorption of a third molecule, which does not interact with the cation (Supplementary Fig. 14) giving rise to a much lower binding energy ( $-42.0$ kJ mol$^{-1}$ ), which suggests that the third molecule is not chemisorbed but physisorbed.

Adsorption on the **1@KOH** structure proceeds somewhat differently. The most stable configuration for the first $SO_2$ molecule adsorbed by **1@KOH** is a monodentate interaction through the one oxygen atom with the extra-framework $K^+$ cation exhibiting a $K \cdots O$ distance of 2.6 Å (see Fig. 5b). This interaction is complemented by hydrogen bonding interaction with a hydroxide group of the metal cluster ($O \cdots H$ distance of 2.1 Å). The second $SO_2$ molecule adsorbed at the crystal defect site does not interact with either the $K^+$ cation or the first molecule. The configuration of the second molecule is characterized by an $AA \cdots DD$ hydrogen bonding pattern between the oxygen atoms of the $SO_2$ molecule and two adjacent

hydroxide groups located at the pyrazolate vacancy sites of the metal cluster (Fig. 5d). The binding energies for the first and second adsorbed molecules are $-88.1$ and $-65.0$ kJ mol$^{-1}$, respectively, which lie in the range of chemisorption. The interaction of the metal cluster hydroxide ions with the $SO_2$ molecules is in agreement with the experimental findings in MTM-300 system[6,28]. The incorporation of a third $SO_2$ molecule gives rise to a weaker interaction with the extra-framework cation and the second adsorbed $SO_2$ molecule with a binding energy of $-44.3$ kJ mol$^{-1}$ characteristic of physisorption (Supplementary Fig. 14).

Therefore, our calculations indicate that **1@Ba(OH)$_2$** has stronger interactions with $SO_2$ molecules than **1@KOH** because of the specific interactions with extra-framework barium cations. The $SO_2$ defect site fixation mechanism parallels the Rubisco type $CO_2$ insertion in the amine–metal coordination bond described by Long et al.[5] on the alkylamine functionalized $M_2$(dobpdc) systems, which implies a cooperative interaction with metal and basic amine sites. Moreover, while the binding energy for the first $SO_2$ molecule falls clearly in the chemisorption range, and the third adsorbed $SO_2$ molecule in the physisorption range, the binding energy for the second molecule might be considered to be between the chemisorption and physisorption ranges. In this regard, the higher interaction of the second adsorbed $SO_2$ molecule with **1@Ba(OH)$_2$** compared with **1@KOH** could explain the fundamental role of the ion exchange process on the enhanced reversible $SO_2$ adsorption.

## Discussion

Here, we show a novel and simple strategy to fine-tune the functional properties of metal organic frameworks towards the directed capture of harmful gases by the deliberate creation of defects, ultimately allowing ion exchange processes. The enhanced adsorptive properties (particularly on the selective $SO_2$ adsorption) of the postsynthetically modified MOF materials series can be taken as a proof of concept of the success of defect engineering methodology for optimizing the adsorptive properties of high connectivity networks. Indeed, the strong and selective interaction of the pore structure with the $SO_2$ target molecule can be attributed to a series of favourable features which include: (i) enhanced basicity of metal hydroxide clusters as a consequence of additional hydroxide anions replacing the missing linkers defects; (ii) affinity of extra-framework $Ba^{2+}$ ions for $SO_2$ fixation; (iii) higher pore accessibility of the 3D structure arising from missing linker defects and (iv) the fine-tuning of the pore surface polarity by the benzene functional groups.

Moreover, the lack of reactivity of co-crystallized barium hydroxide microparticles for $SO_2$ fixation, compared with the high reactivity of extra-framework barium cations in the nickel pyrazolate MOF systems, illustrates the importance of the accessibility and high dispersion of adsorption sites that the defective MOF platform provides.

Future work, to further modulate the adsorptive and catalytic properties of these materials, will fully exploit the ion exchange possibilities opened by the deliberate introduction of defects.

## Methods

**General methods.** Detailed procedures for the synthesis and characterization of ligands and MOF, adsorption of $SO_2$ measurements and DFT calculations are reported in Supplementary Methods.

**Synthesis of materials 1–3.** The synthesis of MOFs samples **1**–**3** were prepared according to the procedure reported by our group[34], with subtle changes as follows: in a typical synthesis, 631 mg (3 mmol) of 4,4'-benzene-1,4-diylbis(1H-pyrazole) were dissolved in 160 ml of N,N'-dimethylformamide and 992 mg (4 mmol) of Ni($CH_3COO$)$_2$ 4 $H_2O$ were dissolved in 40 ml of $H_2O$. The two solutions were mixed and refluxed for 12 h under stirring. The solid obtained was filtered off and

washed with N,N'-dimethylformamide, ethanol and diethyl ether, yielding the corresponding MOF *1–3*. Before use or characterization of materials, 500 mg of as synthesized solids were solvent exchanged with 100 ml of dichloromethane, with stirring at room temperature for 2 h.

**Preparation of materials 1@KOH—3@KOH.** The postsynthetical modification of **1**, **2** and **3** materials were done according to previously reported procedure by our group[20], with activation of as synthesized MOFs thermally at 423 K and outgassed to $10^{-1}$ Pa for 12 h, to obtain solvent-free porous matrix. Afterwards, 0.055 mmol of each material was suspended in 0.35 M KOH absolute ethanol solution (5.5 ml). The resulting suspensions were stirred overnight under an inert $N_2$ atmosphere, filtered off and washed copiously with absolute ethanol yielding the corresponding compounds **1@KOH**, **2@KOH**, **3@KOH**.

**Preparation of materials 1@Ba(OH)$_2$—3@Ba(OH)$_2$.** The materials **1@KOH**, **2@KOH**, **3@KOH** were used as prepared without previous activation as follow, 100 mg of the **1@KOH–3@KOH** materials were suspended in 12 ml of a 0.1 M aqueous solution of the Ba(NO$_3$) with stirring for 72 h at room temperature. The postmodified materials were subsequently filtered off, washed with water and ethanol and dried in air. Later, the solids (~50 mg) as obtained were suspended in 50 ml of water for 4 h to remove the eventual absorbed ion pairs. The materials **1@Ba(OH)$_2$**, **2@Ba(OH)$_2$** and **3@Ba(OH)$_2$** were filtered off and washed with water and ethanol, and dried in air.

Postsynthethical modified material **1@Ba(OH)$_2$**: Elemental analysis (previously activated sample) Calculated for Ba$_{0.5}$(Ni$_8$(OH)$_3$(C$_2$H$_5$O)$_3$(H$_2$O)$_2$(C$_{12}$H$_8$N$_4$)$_{5.5}$) (Ba(OH)$_2$)(H$_2$O)$_5$.% C,39.9; N, 14.22; H, 3.63; Found, C,40.05; N,14.32; H, 3.90. Calculated residual oxides from TGA for **1@Ba(OH)$_2$**: (NiO)$_8$(BaO)$_{1.5}$ 38.18%; Found: 38.26%. ICP-MS composition for **1@Ba(OH)$_2$**: Ni, 5.75 p.p.m.; Ba, 2.82 p.p.m. ATR-FTIR (4,000–400 cm$^{-1}$): 3,585(br), 3,373(br), 3,141(br), 3,023(w), 2,967(w), 1,578(s), 1,453(vs), 1,356(m), 1,246(m), 1,164(w), 1,123(w), 1,048(s), 956(s), 820(vs), 651(w)532(w),503(w).

Postsynthethical modified material **2@Ba(OH)$_2$**: Elemental analysis (previously activated sample) Calculated for Ba$_{1.5}$(Ni$_8$(OH)$_3$(C$_2$H$_5$O)(H$_2$O)$_2$(C$_{12}$H$_7$N$_4$O)$_5$) (Ba(OH)$_2$)$_{0.5}$(H$_2$O).% C,36.73; N, 13.81; H, 2.48; Found: C,36.88; N,13.46; H, 3.05. Calculated residual oxides from TGA for **2@Ba(OH)$_2$**: (NiO)$_8$(BaO)$_{4.5}$%; Found: 43.83%. ICP-MS composition for **2@Ba(OH)$_2$**: Ni, 6.90 p.p.m.; Ba, 4.10 p.p.m. ATR-FTIR (4,000–400 cm$^{-1}$): 3,388(br), 3,064(w), 2,975(w), 1,623(m), 1,573(s), 1,456(vs), 1,371(m), 1,249(m), 1,211(w), 1,170(w), 1,058(s), 958(m), 856(vs), 823(s), 692(w), 665(w),613(w),567(w).

Postsynthethical modified material **3@Ba(OH)$_2$**: Elemental analysis (previously activated sample) Calculated for Ba$_{0.5}$(Ni$_8$(OH)$_3$(C$_2$H$_5$O)$_3$(H$_2$O)$_2$(C$_{12}$H$_9$N$_5$)$_{5.5}$) (Ba(OH)$_2$)$_{1.5}$(H$_2$O).% C,38.20; N, 17.02; H, 3.41; Found: C,38.11; N,17.02; H, 3.83. Calculated residual oxides from TGA for **3@Ba(OH)$_2$**: (NiO)$_8$(BaO)$_2$ 39.95%; Found: 40.15%. ICP-MS composition for **3@Ba(OH)$_2$**: Ni, 6.11 p.p.m.; Ba, 3.29 p.p.m. ATR-FTIR (4,000–400 cm$^{-1}$): 3,583(w), 3,365(br), 3,045(w), 2,970(w), 1,622(m), 1,572(s), 1,460(vs), 1,384(m), 1,246(s), 1,168(m), 1,126(w), 1,058(vs), 958(m), 856(m), 810(vs), 665(w), 617(w), 563(w), 501(w), 474(w).

**Adsorption isotherm of SO$_2$ on 1@Ba at 303 K.** Adsorption isotherm was measured at 303 K, point by point using breakthrough experiments with total flow of 30 ml min$^{-1}$ He/SO$_2$ variable gas mixtures from (97.5/2.5) to (25/75). The MOFs is activated at 423 K for 24 h before first chemisorption cycle, and for 2 h between further cycles. Each gas mixture is measured twice to assure the physisorption process. The adsorbed amount was calculated using the same procedure as for routine breakthrough experiments. It should be noted that all isotherm points are measured on the same prepared column and the material characterized after this adsorption cycles maintain crystallinity and porosity. The desorption branch was unable to measure with this procedure.

**Breakthrough experiments for gas separation.** For these measurements, the PSM materials used were carefully handled avoiding possible chemisorption of CO$_2$ from air and the 20-cm chromatographic column, that was prepared employing a stainless steel 20 cm-column (0.4 cm internal diameter) packed with ca. 0.5 g of the studied materials (**1–3**, **1@KOH–3@KOH** and **1@Ba(OH)$_2$–3@Ba(OH)$_2$**. The column was activated under a pure He flow (20 ml min$^{-1}$) at 423 K overnight and for 2 h between successive breakthrough cycles. The desired gas mixture (20 ml min$^{-1}$) was prepared via mass flow controllers. For instance, N$_2$/SO$_2$ (97.5: 2.5), N$_2$/CO$_2$/SO$_2$ (83.5: 14: 2.5) and N$_2$/H$_2$O/SO$_2$ (94.1: 3.4: 2.5) gas mixtures were prepared to simulate the emission of flue gas from a power plant. The breakthrough experiments were carried out, at 303 K, by step changes from He to N$_2$/SO$_2$, N$_2$/CO$_2$/SO$_2$, and N$_2$ /H$_2$O/SO$_2$ flow mixtures. The subsequent breakthrough cycles were measured with prior sample reactivation under a pure He flow (20 ml min$^{-1}$) at 423 K during 2 h. The relative amounts of gases passing through the column were monitored on a Mass Spectrometer Gas Analysis System (Pfeiffer Vacoon) detecting ion peaks at $m/z$ 64 (SO$_2$), 44 (CO$_2$), 28 (N$_2$), 18 (H$_2$O) and 4 (He). The adsorbed amounts of SO$_2$ for the different materials are summarized in Supplementary Table 3.

**Variable temperature pulse gas chromatography.** Gas-phase adsorption at zero-coverage surface was studied using the pulse chromatographic technique[35] employing a gas chromatograph and stainless steel 20 cm-column (0.4 cm internal diameter) packed with ca. 0.5 g of the studied materials (**1–3**, **1@KOH–3@KOH** and **1@Ba(OH)$_2$–3@Ba(OH)$_2$**). It should be noted that all the measurements were done on the materials after SO$_2$ chemisorption (after breakthrough experiments) to ensure the thermodynamic equilibrium. Before measurement, samples were heated overnight at 423 K in a He flow (20 ml min$^{-1}$). Later on, an equimolecular gas mixture composed of H$_2$, N$_2$, CO$_2$, SO$_2$ gases (0.4 ml) was injected at 1 bar and the separation performance of the chromatographic column was examined at different temperatures (403–433 K) by means of a mass Spectrometer Gas Analysis System (Pfeiffer Vacoon), detecting the corresponding masses. The dead volume of the system was calculated using the retention time of hydrogen as a reference. The zero-coverage thermodynamic parameters of the adsorption process for SO$_2$ and CO$_2$ are gathered in Table 1 and Supplementary Table 4, respectively. These values were calculated using a van't Hoff type analysis employing isothermal chromatographic measurements[5]. The retention volumes were corrected taking into account the volume expansion of the gas entering the capillary due to the temperature increase according to $V_S = (t_R - t_m)F_a(T/T_a)j$ where $V_S$ = net retention volume (ml); $t_R$ = retention time ($_{min}$); $t_m$ = dead time ($_{min}$); $F_a$ = volumetric flow-rate measures at ambient temperature (ml min$^{-1}$); $T$ = column temperature (K); $T_a$ = ambient temperature (K); the James–Martin gas compressibility correction $j = (3(p_i/p_0)^2 - 1)/(2(p_i/p_0)^3 - 1)$ where $p_i$ = pressure of gas applied to the chromatogram and $p_0$ = pressure of gas at outlet.

Once these corrections where applied, the van't Hoff plot of the equation $\ln V_S = \ln(RTn_s) + \Delta S/R - \Delta H_{diff}/(RT)$ was used to calculate the thermodynamic parameters of each analyte taking into account that the term $\ln(RTn_s)$ is usually small and can be neglected in the determination of $\Delta S$. In addition to the $\Delta H_{diff}$ value obtained from the van't Hoff plot the isosteric heat of adsorption ($\Delta H_{iso}$) was also determined according to the relation $\Delta|H_{iso}| = |\Delta H_{diff}| + RT_{average}$. The $\alpha_{SO_2/CO_2}$ partition coefficients have been calculated from the Henry constants ratio according to the relation the $\alpha_{SO_2/CO_2} = K_{H\_SO_2}/K_{H\_CO_2} = \exp[-(\Delta G_{SO_2} - \Delta G_{CO_2})/RT]$.

**Characterization of materials after SO$_2$ chemisorption.** The materials were characterised after SO$_2$ chemisorption process by means of EA, XRPD, FTIR, TEM-EDX, TGA-FTIR and N$_2$ adsorption isotherms to know the effect of the chemisorption process on the structural integrity of the material. The results are indicative that the crystal phase of both MOF and Ba(OH)$_2$ cocrystals are maintained and only a slight diminution on surface area is observed as a probable consequence of the formation of BaSO$_3$ nanoclusters.

**Elemental analysis after adsorption of SO$_2$.** **1@Ba(OH)$_2$** after adsorption of SO$_2$: elemental analysis: found % C,36.81; N,13.45; H, 3.56; S, 0.29. Calculated for Ba$_{0.5}$(Ni$_8$(OH)$_4$(C$_2$H$_5$O)$_2$(C$_{12}$H$_8$N$_4$)$_{5.5}$)(Ba(OH)$_2$) (H$_2$O)$_{14}$ (SO$_2$)$_{0.2}$ C,36.9; N,13.52; H, 3.89; S, 0.28. ATR-FTIR (4,000–400 cm$^{-1}$): 3,360(br), 3,028(w), 1,578(vs), 1,464(vs), 1,385(w), 1,355(s), 1,247(vs), 1,166(m), 1,145(w), 1,124(m), 1,047(s), 955(s), 822(vs), 857(w), 651(w), 677(w), 649(m), 534(m), 503(m).

**2@Ba(OH)$_2$** after adsorption of SO$_2$: elemental analysis: found C,29.72; N, 10.91; H, 3.51; S, 0.59; calculated for Ba$_{1.5}$(Ni$_8$(OH)$_3$(C$_2$H$_5$O)(H$_2$O)$_2$(C$_{12}$H$_7$N$_4$O)$_5$)(Ba(OH)$_2$)$_{0.5}$(H$_2$O)$_{27}$ (SO$_2$)$_{0.5}$ C,29.46; N, 11.08; H, 4.08; S, 0.63. ATR-FTIR (4,000–400 cm$^{-1}$): 3,385(br), 1,625(m), 1,578(s), 1,459(vs), 1,375(w), 1,249(m), 1,201(w), 1,188(w), 1,088(w), 1,057(s), 957(w), 943(w), 856(s), 814(s), 693(w), 673(w), 611(m),571(w), 557(w).

**3@Ba(OH)$_2$** after adsorption of SO$_2$: elemental analysis: found: C, 33.46; N, 14.14; H, 3.42; S, 0.46; calculated for Ba$_{0.5}$(Ni$_8$(OH)(C$_2$H$_5$O)$_5$(H$_2$O)$_2$(C$_{12}$H$_9$N$_5$)$_{5.5}$) (Ba(OH)$_2$)$_{1.5}$(H$_2$O)$_{20}$(SO$_2$)$_{0.4}$ C, 33.96; N,14.33; H, 4.59; S, 0.47. ATR-FTIR (4,000–400 cm$^{-1}$): 3,580(w), 3,380(br), 1,624(m), 1,573(s), 1,454(vs), 1,386(m), 1,373(w), 1,246(s), 1,172(m), 1,122(w), 1,108(w), 1,060(vs), 985(m), 946(w), 856(s), 822(s), 693(w), 668(w), 613(s), 571(w).

**Computational details of periodic DFT calculations.** A theoretical study of **1**, **1@KOH**, and **1@Ba(OH)$_2$** was carried out using DFT, as implemented in the VASP programme[22]. The calculations were performed with a cutoff energy of 500 eV and PAW potentials[36]. The PBE exchange-correlation functional[37], with corrected van der Waals interactions introduced via the D2 Grimme scheme[38] was used. Due to the large sizes of the unit cells, only the gamma point was used. No symmetry constrains were used and both the atomic coordinates and the cell parameters were allowed to vary.

Calculations on **1**, **1@KOH**, **1@Ba(OH)$_2$** were performed with a cubic primitive cell containing 166 and 306 framework atoms for **1**, and for **1@KOH** and **1@Ba(OH)$_2$**, respectively. The initial cubic cell has the cell parameters equal to $a = 25.38$ Å. Due to the presence of local defects, the optimized cells depart from the ideally cubic structure, as it is common in defective porous materials[39]. This is not a problem in the present study, as our interest lies on the local host–guest interactions. What it is important is the accurate description of the local structure and the host–guest interactions. Note that indeed both bond lengths and angles are in agreement with reported crystal data.

**Computational study of the adsorption of SO₂ molecules.** The initially high symmetry of the material allows us to model the defective solid with a relative small number of configurations. In this context, three different configurations were considered for the extraction of the missing linkers and the introduction of $K^+$ cations. The dangling metal-N bonds were capped with OH. In the case of the replacement of two $K^+$ cations by one $Ba^{2+}$ one, four configurations were taken into account. As mentioned above, the local structure is strongly affected by the presence of defects, as can be seen in Supplementary Fig. 14 and Supplementary Table 6. It is observed that larger distortions result from the exchange of $K^+$ by $Ba^{2+}$ cations, as a consequence of the two times larger polarizing power of $Ba^{2+}$.

**Data availability.** Details of the experimental and computational methods to characterize the materials and their adsorptive properties are included in this published article and its Supplementary Information file or upon request to the corresponding authors.

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

## Acknowledgements

We thank generous funding from the European Research Council through an ERC Starting Grant (ERC2011-StG-279520-RASPA), the Spanish Ministry of Economy (CTQ2013-48396-P, CTQ2014-53486-R, CTQ2015-70135-REDT) and FEDER and Marie Curie IIF-625939 (LMRA) funding from European Union and Andalucía Region (FQM-1851). We would also like to thank the high performance computer Centre Alhambra (at the University of Granada), and the Centro Informático Científico de Andalucía (CICA), for providing us computer resources to carry out the work.

## Author contributions

L.M.R.-A. and E.L.-M. carried out the synthesis and full characterization of the MOF materials. S.H., A.R.R.-S. and S.C. carried out the computational simulation work. L.M.R.-A. and J.A.R.N. analyzed the experimental data and wrote the manuscript with collaboration of S.H. and A.R.R.-S. J.A.R.N. was responsible for the overall direction of the Project with contributions from all the authors.

## Additional information

**Competing financial interests:** The authors declare no competing financial interests.

