## [Peer Review File · Nature Communications]

Reviewers' Comments:

Reviewer #1 (Remarks to the Author)

The manuscript by J. Navarro et al. describes one of the first examples of the use of a Metal Organic Frameworks for the selective adsorption of sulfur dioxide. In order to tune MOF affinity for this adsorbate, the authors introduce a two step post-synthetic method consisting on the creation of defects in nickel pyrazolate type MOFs followed by the introduction of Ba that seems to form nano/micro crystals of $\text{Ba}(\text{OH})_2$ within the MOF porosity. Not surprisingly, materials containing Ba display a quite impressive capacity for the capture of SO_2 (circa 5.6 mmol/g).

After reading the manuscript with interest, I do believe that these results, and specially the sequential functionalization of this interesting family of MOFs may become of interest for readers of Nature Communications and I do support its publication. I have however a series of points that I think the authors should address before the paper can be accepted:

- The main point is that the manuscript is, at some points (most of it actually but specially the discussion in page 3 and beginning of page 4) quite tedious and difficult to read. Not all the information here presented is relevant for the topic at hand and could be moved to the SI in order to make the text more amenable (i.e. what is the error in the calculation of the Gibbs free energy for the different solids? Is a difference of 1kJ/mol really meaningful and worth to be described here?), when discussing breakthrough experiments, the regeneration protocol is not given, making therefore very difficult for the reader...
- Nature of irreversible adsorbed SO_2 : from the discussion, it seems that only Ba present as extra cationic species within the framework is responsible for this strong adsorption of SO_2 , however, if this Ba^{2+} is transformed into BaSO_4 , then what happens with the charge compensation?
- Periodic DFT calculations: although I am far from being an expert, I can imagine that defects are not something easy to simulate, indeed quite a number of research groups are now busy trying to develop more realistic models. What is new in this work and how realistic are these simulations?

Reviewer #2 (Remarks to the Author)

The manuscript describes the adsorption and separation of SO_2 from CO_2 and N_2 by defect MOF materials based upon a Ni(II)-pyrazole coordination. The treatment of the parent MOFs with metal hydroxide (KOH , $\text{Ba}(\text{OH})_2$) gives defect materials that have been previously reported (in ref 22 and 23), and these have been used successfully to study selective uptake and breakthrough experiments with the above gases. This is interesting work especially since relatively little work has been conducted on SO_2 adsorption and selectivity with the vast majority of coordination compounds (MOFs) are unstable and react with SO_2 .

As can be seen in Table 2 though, there are clear reductions in SO_2 capacity from cycle 1 to 3 in all cases reflecting some decomposition, pore collapse or reactivity of the material. I searched the files extensively and could not find any isotherm plots for SO_2 uptake in these materials, only N_2 uptake post SO_2 uptake. This is a serious omission since the SO_2 isotherm is the clearest and obvious way of assessing the degree of reversibility of SO_2 uptake and release over multiple cycles. This should be added to the manuscript, and without this the manuscript is not publishable in its current form. I suspect that the materials are not stable to SO_2 (based upon the data Table 2) and although there are some nice ideas here I would recommend major revision subject to inclusion of the SO_2 isotherms over multiple cycles.

The manuscript is not an easy read and is very condensed in parts.

Reviewer #3 (Remarks to the Author)

This manuscript describes a defect engineering strategy for enhancing the SO₂ adsorption capacity and selectivity of a Ni-containing MOF. The use of intentionally introduced defects to tune the SO₂ adsorption is quite novel and creative and represents a potential new paradigm of MOF engineering / optimization. The results obtained for SO₂ adsorption are also fairly impressive, particularly at the low (relevant) partial pressures. As such, I would expect that this manuscript will be of fairly wide and general interest to the large MOF community. While the manuscript is generally seems well written and executed, there are several minor concerns that should be addressed prior to publication.

In particular, the authors state that "These [breakthrough] experiments were carried out on the materials after the SO₂ chemisorption process (see below)..." This is really never clarified later in the paper. Given that some amount of the SO₂ adsorption is irreversible, it is important to define what type of treatment/exposure that the MOF underwent prior to the breakthrough measurements.

The authors also complement their experimental measurements with some DFT calculations of SO₂ binding to various sites. The minimum energy binding sites are very stable (~100 kJ/mol). This is far stronger than suggested by the analysis of the chromatography. Rather, the calculated secondary binding site energy is much similar to that obtained from chromatography. Is the premise that the strong (Ba²⁺ bound) sites are irreversibly bound -- thus no longer contributing in the equilibrium regime -- with the OH sites serving as the reversible adsorption sites? If so, this should be stated explicitly. Furthermore, it should be possible to provide an estimate of the amount of SO₂ bound on the Ba²⁺ vs. the OH- via a site counting argument. Does this fraction roughly correspond to the fraction of irreversibly bound SO₂? It would also be interesting to attempt to probe the binding energy of the "irreversibly" bound SO₂, either at high temperatures or via calorimetric measurements on virgin (never exposed to SO₂) MOF.

In short, the only major omission from this manuscript is an explicit hypothesis regarding the molecular-level mechanism for irreversible vs. reversible SO₂ adsorption, and associated data to back up this hypothesis. While a hypothesis seems to be implied, it is up to the reader to infer it, and just a bit more supporting data would help nail it down.

Dear Editor,

We have modified the manuscript thoroughly in order to improve the scholar presentation and consequently we have not highlighted the changes. Moreover, in order to address the reviewers comments, in full, we have carried out additional experiments (TGA with FT-IR analysis of evolved gases, SO₂ adsorption isotherm) as well as additional DFT calculations. Details of the changes and answer to comments of the reviewers are given below:

Reviewer 1:

Reviewer comment:

- The main point is that the manuscript is, at some points (most of it actually but specially the discussion in page 3 and beginning of page 4) quite tedious and difficult to read. Not all the information here presented is relevant for the topic at hand and could be moved to the SI in order to make the text more amenable (i.e. what is the error in the calculation of the Gibbs free energy for the different solids? Is a difference of 1kJ/mol really meaningful and worth to be described here?), when discussing breakthrough experiments, the regeneration protocol is not given, making therefore very difficult for the reader.

Answer: We have improved the discussion of pages 3 and 4 in order to make it more scholarly with the details of reactivation in the successive breakthrough curve being clearly given in the text.

The 1kJ/mol difference in Gibbs free energy is not meaningful with error being > 4kJ/mol and consequently the comment has been erased from the discussion.

Reviewer comment:

- Nature of irreversible adsorbed SO₂: from the discussion, it seems that only Ba present as extra cationic species within the framework is responsible for this strong adsorption of SO₂, however, if this Ba²⁺ is transformed into BaSO₄, then what happens with the charge compensation?

Answer: Ba is not the only source of SO₂ fixation but there is a cooperative effect of Ba and metal cluster hydroxide ions. Charge compensation arises from hydroxide ions consumption from metal hydroxide clusters with concomitant formation of BaSO₃ and ulterior oxidation to BaSO₄ under open atmosphere conditions. We have clarified this point by adding equations 1 and 2 to the discussion (page 5).

Reviewer comment:

- Periodic DFT calculations: although I am far from being an expert, I can imagine that defects are not something easy to simulate, indeed quite a number of research groups are now busy trying to develop more realistic models. What is new in this work and how realistic are these simulations?

Answer: We fully agree with the reviewer that modelling of defects is not routine work. In this regard, we have added a paragraph highlighting the difficulty DFT modeling of defective metal organic frameworks with some recent examples found in the literature of (see pages 3, 7 and 8). Moreover, in order to have realistic models of defective framework geometries a large variety of configurations have been explored in order to find an optimized structure (see section S5 of supporting information for more details).

Reviewer 2:

Reviewer comment:

As can be seen in Table 2 though, there are clear reductions in SO₂ capacity from cycle 1 to 3 in all cases reflecting some decomposition, pore collapse or reactivity of the material. I searched the files extensively and could not find any isotherm plots for SO₂ uptake in these materials, only N₂ uptake post SO₂ uptake. This is a serious omission since the SO₂ isotherm is the clearest and obvious way of assessing the degree of reversibility of SO₂ uptake and release over multiple cycles. This should be added to the manuscript, and without this the manuscript is not publishable in its current form. I suspect that the materials are not stable to SO₂ (based upon the data Table 2) and although there are some nice ideas here I would recommend major revision subject to inclusion of the SO₂ isotherms over multiple cycles.

Answer:

1. We have measured a SO₂ adsorption isotherm for the representative material 1@Ba(OH)₂ (SI-Fig.S8) using breakthrough measurements of He:SO₂ gas mixtures with the material. Noteworthy, the material maintains its integrity after several adsorption desorption cycles (details are given in the supporting information section S3).
2. The irreversible SO₂ chemisorption process in the defective materials containing defects is now clarified on the basis of increased basicity and presence of extraframework cations according to equations 1 and 2 (page 5). The characterization of the materials after SO₂ chemisorption is indicative of maintenance of framework crystallinity (XRPD patterns after SO₂ adsorption are now included in the Figure 2b). Moreover, a small decrease of pore accessibility to N₂ (diminution of ca. 10%) is also noticed as a probable consequence of formation of BaSO₃ nanoclusters in the pore voids. We have also carried out a thermogravimetric analysis, under nitrogen atmosphere, of the 1@Ba(OH)₂ material with chemisorbed SO₂ with FTIR study of evolved gases. The results are indicative of chemisorbed SO₂ evolution in the 511 to 594 K temperature range, arising from sulphite decomposition, just before starting the framework pyrolysis (see page 5 and SI-Sect.4 Fig. S11). Noteworthy, ex situ XRPD pattern of material heated at 603 K is indicative of maintenance of MOF crystallinity (see discussion of page 5).

Comment: The manuscript is not an easy read and is very condensed in parts.

Answer: The manuscript text has been fully revised in order to improve its scholarly presentation.

Reviewer #3

Reviewer comment:

The authors state that "These [breakthrough] experiments were carried out on the materials after the SO₂ chemisorption process (see below)..." This is really never clarified later in the paper. Given that some amount of the SO₂ adsorption is irreversible, it is important to define what type of treatment/exposure that the MOF underwent prior to the breakthrough measurements.

Answer: We agree with the reviewer that the original discussion on dynamic adsorption of SO₂ paragraph of pages 3-4 was confusing. In order to clarify the discussion we have changed the sequence of the discussion of results. Now, we start discussing the breakthrough curve experiments which were carried out on the fresh materials (prior to SO₂ adsorption). It is important to note that all materials were activated at 423K, in helium flow (20 mL min⁻¹) for 24 hours prior to the first adsorption cycle and for 2 hours between the successive breakthrough cycles. The discussion is followed now by the variable temperature pulse gas chromatographic studies. It should be noted that in this case the materials used were the ones after SO₂ adsorption since pulse gas measurements requires thermodynamic equilibrium conditions (reversible adsorption).

Reviewer comment:

The authors also complement their experimental measurements with some DFT calculations of SO₂ binding to various sites. The minimum energy binding sites are very stable (~100 kJ/mol). This is far stronger than suggested by the analysis of the chromatography. Rather, the calculated secondary binding site energy is much similar to that obtained from chromatography. Is the premise that the strong (Ba²⁺ bound) sites are irreversibly bound -- thus no longer contributing in the equilibrium regime -- with the OH sites serving as the reversible adsorption sites? If so, this should be stated explicitly.

Answer: We have carried out additional DFT calculations which include now the interaction of the defect sites with up to three SO₂ molecules. The results show that the two first adsorbed molecules give rise to high binding energies (chemisorption process) originating from interactions of SO₂ molecules with extraframework cations and hydroxide metal cluster replacing missing linker vacancies. By contrast the third adsorbed molecule gives rise to lower energies, namely in the physisorption range (40 kJ mol⁻¹) agreeing with values found experimentally by gas chromatography experiments (see discussion of pages 7 and 8 and section S5 of SI).

Reviewer comment: Furthermore, it should be possible to provide an estimate of the amount of SO₂ bound on the Ba²⁺ vs. the OH⁻ via a site counting argument. Does this fraction roughly correspond to the fraction of irreversibly bound SO₂?

Answer: The analysis of samples after chemisorption are indicative of a 0.2-0.6 moles of SO₂ per metal cluster. This corresponds to aprox. 0.5 BaSO₃ nanoclusters per formula unit agreeing with the content of extraframework barium content in the materials.

Reviewer comment: It would also be interesting to attempt to probe the binding energy of the "irreversibly" bound SO₂, either at high temperatures or via calorimetric measurements on virgin (never exposed to SO₂) MOF.

Answer: We have now measured TGA analysis with study of the composition of evolved gases on the 1@Ba(OH)₂ after exposition to SO₂. The FT-IR analysis of evolved gases during thermo gravimetric analysis experiments, under an inert atmosphere of N₂, are indicative of SO₂ evolution, in the 511 to 594 K temperature range, arising from sulphite decomposition just before starting the framework pyrolysis (see page 5 and SI-Sect.4 Fig. S11).

Reviewer comment:

In short, the only major omission from this manuscript is a explicit hypothesis regarding the molecular-level mechanism for irreversible vs. reversible SO₂ adsorption, and associated data to back up this hypothesis. While a hypothesis seems to be implied, it is up to the reader to infer it, and just a bit more supporting data would help nail it down.

Answer: The mechanism of irreversible SO₂ binding has been clarified in page 4 (equations 1 and 2) as well as the DFT calculations discussion of page 7 and 8. Regarding the reversible binding we believe that there is not only one factor but a probable mixture of them (improved accessibility after creation of defects, increased basicity, charge gradients of extraframework cations, polar tags in organic linkers, etc) as discussed in the dynamic gas adsorption section (page 5) and DFT calculations (pages 7-8).

We hope that the new version of the manuscript meets the standard of publication in Nature Communications.

Kind regards,

Marleny Rodriguez Albelo

Jorge A. R. Navarro

Reviewers' Comments:

Reviewer #1 (Remarks to the Author)

I have read the response to the referee comments and the new version of this manuscript. I have the strong opinion that the paper has improved substantially and that the most pressing comments from all referees were properly addressed by the authors, therefore I do suggest acceptance of the work in its current form.

Reviewer #2 (Remarks to the Author)

I find this a tortuous manuscript to read. It is very condensed with some language issues coupled to some complex descriptions of inter-conversions and analyses.

I requested that SO₂ isotherms over multiple cycles should be added. Fig S8 shows an uptake plot for SO₂ but no desorption and no further cycles. I can only assume therefore that the uptake is essentially irreversible via formation of BaSO₃ within and/or outside the MOF.

The formation of defect structures with enhanced properties has been reported by other authors, and I am left wondering what conceptually I have learnt from this manuscript that I did not know before reading it. It is certainly a nice combination of post-synthetic modification via ligand loss coupled to cation exchange processes (Fig 1). The properties of the resultant defect material are "better" than the parent material, but I am left wondering how this is conceptually new. To my knowledge this is new SO₂ chemistry within a MOF environment, but really this is also just a reaction of BaO with SO₂ to give BaSO₃.

On balance I do not feel that the manuscript gives a clearly defined high impact message and regrettably do not recommend acceptance.

Reviewer #3 (Remarks to the Author)

The revised manuscript has addressed most of my prior concerns. The revised manuscript is significantly more complete and readable. That said, I do have several additional (new) concerns regarding the revised manuscript that should be addressed prior to publication.

In several places (most prominently, in the abstract) the authors refer to the MOF as "robust". It is true that some evidence is presented that the structure does not immediately degrade in the presence of SO₂. But a longer term exposure is necessary before declaring the structure "robust". In any industrially relevant usage, "robust" would mean months or years of exposure! While not essential for publication, one should be careful of the claim. Furthermore, the fact that the MOF undergoes irreversible SO₂ uptake also seems to indicate a lack of robustness in terms of its SO₂ capture behavior.

The authors now specifically comment on the reversibility of the SO₂ adsorption. This is nicely outlined in Figure 3. However, the authors still do not provide a complete explanation for the role of the exchanged cations, particularly in terms of the REVERSIBLE uptake. In particular, the authors seem to argue that the SO₂ molecules that directly interact with the metal cation are strongly adsorbed and form the "irreversible" chemisorbed component of the uptake. The remaining (reversibly adsorbed) SO₂ do not interact with the cation and are more weakly physisorbed. But what is the proposed role of the cations in this latter reversible component? Why bother doing cation exchange at all, since (in practice) it is only the reversible component that is useful? Certainly the DFT results do not seem to indicate enhanced physisorption of the "outer sphere" SO₂ molecules. Note, however, that it does appear that the "reversible" component also

benefits from cation exchange as well, although no explanation for this observation is given.

Along the same lines, the TPD measurements (Figure 4) show very little difference in the SO₂ adsorption enthalpies for the K⁺ vs. Ba²⁺ exchanged MOF. Since these TPD were done at "steady state", they should be probing the physisorbed SO₂. Consistent with my prior discussion, these measurements also suggest that one should expect little difference in the useful, reversible, working capacities.

If the authors are able to provide a more compelling explanation of these key observations (Ba²⁺ vs. K⁺ enhancement), it would greatly strengthen the manuscript.

Minor issues:

* On page 2, "being barium" should read "barium being"

Response to the comments of Reviewer 2:

“I find this a tortuous manuscript to read. It is very condensed with some language issues coupled to some complex descriptions of inter-conversions and analyses.”

Response: The manuscript has been carefully revised and edited by a native researcher expert in the field in order to allow a fluid reading.

“I requested that SO₂ isotherms over multiple cycles should be added. Fig S8 shows an uptake plot for SO₂ but no desorption and no further cycles. I can only assume therefore that the uptake is essentially irreversible via formation of BaSO₃ within and/or outside the MOF. The formation of defect structures with enhanced properties has been reported by other authors, and I am left wondering what conceptually I have learnt from this manuscript that I did not know before reading it. It is certainly a nice combination of post-synthetic modification via ligand loss coupled to cation exchange processes (Fig 1). The properties of the resultant defect material are "better" than the parent material, but I am left wondering how this is conceptually new. To my knowledge this is new SO₂ chemistry within a MOF environment, but really this is also just a reaction of BaO with SO₂ to give BaSO₃”.

Response:

It is apparent from reviewers comments that his concerns are related to two main points: (a) the reversibility of the SO₂ adsorption process and (b) the possible lack of conceptual novelty of the interaction of SO₂ molecules with “BaO”.

Clarification to point (a):

1. The isotherm points have been obtained from breakthrough experiments carried out in the same material after variation of the He:SO₂ relative compositions as indicated in SI (Section S3). This means that the material has been activated each time assuring the reversibility of the adsorption process at the different SO₂ partial pressures.
2. We have carried out a new breakthrough experiment in which we have performed 10 successive activation and adsorption cycles under a constant flow of 20 mL min⁻¹ of SO₂:N₂ (2.5:97.5) mixture which is shown in Fig. 3f and discussed in last paragraph of page 4. The

results are indicative of materials stability to SO₂ exposure and of the reproducibility of the adsorption process over multiple cycles.

Clarification to point (b): We fully disagree with reviewer comment. In this regard, we clearly demonstrate the high reactivity of extraframework Ba²⁺ in combination with basic hydroxide metal clusters inside the MOF particles compared to the lack of reactivity of Ba(OH)₂ microcrystals. These results clearly point to the importance of the high dispersion of barium cations and hydroxide anions provided by the defective MOF platform. See discussion of page 5 paragraph 2. Moreover, we have also added a conclusion paragraph (page 9 paragraph 2) highlighting this differentiated reactivity.

2. Response to the comments of Reviewer 3:

“In several places (most prominently, in the abstract) the authors refer to the MOF as "robust". It is true that some evidence is presented that the structure does not immediately degrade in the presence of SO₂. But a longer term exposure is necessary before declaring the structure "robust". In any industrially relevant usage, "robust" would mean months or years of exposure! While not essential for publication, one should be careful of the claim. Furthermore, the fact that the MOF undergoes irreversible SO₂ uptake also seems to indicate a lack of robustness in terms of its SO₂ capture behavior.”

Response: We agree with the reviewer, so the term “robust” have been omitted or changed by synonyms, as we refer to the term as the chemical stability of our materials compared to other porous coordination polymers. Moreover, as above mentioned we have carried out a new breakthrough experiment over 10 successive activation and adsorption cycles under a constant flow of 20 mL min⁻¹ of SO₂:N₂ (2.5:97.5) mixture (see Fig. 3f) and discussed in last paragraph of page 4. The results are indicative of materials stability to SO₂ exposure and of the reproducibility of the adsorption process over multiple cycles.

“The authors now specifically comment on the reversibility of the SO₂ adsorption. This is nicely outlined in Figure 3. However, the authors still do not provide a complete explanation for the role of the exchanged cations, particularly in terms of the REVERSIBLE uptake. In particular, the authors seem to argue that the SO₂ molecules that directly interact with the metal cation are strongly adsorbed and form the "irreversible" chemisorbed component of the uptake. The remaining (reversibly adsorbed) SO₂ do not interact with the cation and are more weakly physisorbed. But what is the proposed role of the cations in this latter reversible component? Why bother doing cation exchange at all, since (in practice) it is only the reversible component that is useful? Certainly the DFT results do not seem to indicate enhanced physisorption of the "outer sphere" SO₂ molecules. Note, however, that it does appear that the "reversible" component also benefits from cation exchange as well, although no explanation for this observation is given.”

“Along the same lines, the TPD measurements (Figure 4) show very little difference in the SO₂ adsorption enthalpies for the K⁺ vs. Ba²⁺ exchanged MOF. Since these TPD were done at "steady state", they should be probing the physisorbed SO₂. Consistent with my prior discussion, these measurements also suggest that one should expect little difference in the useful, reversible, working capacities.”

Response: The experimental results of reversible SO₂ adsorption (working capacities) clearly point to a benefit of ion exchange process. In this regard, the binding energy for the adsorbed second molecule of SO₂ can be considered to lie between chemisorption and physisorption processes. Noteworthy, the binding energy for the barium exchanged material is higher than the potassium system which points to the beneficial impact of the ion exchange process. In order to clarify this point we have added a phrase in page 8 paragraph 3.

Reviewers' Comments:

Reviewer #2 (Remarks to the Author)

I requested that isotherms for SO₂ uptake over multiple cycles be included. This required extending the experiment already shown in Figure S8 (which has a strange shape actually) to show multiple sorption - cycles desorption so that it is absolutely clear and unambiguous what the level of reversibility is in this system. The authors have chosen not to do this but show breakthrough cycles instead. This still raises the confusion in my mind as to the levels of chemi- and physi-sorbed SO₂ in these materials and what is the behaviour of the material in terms of formation of BaSO₃ inside or outside the MOF pores. The new experiments are conducted on N₂/SO₂(97.5:2.5) flow (ie highly diluted SO₂ streams) and so does not cover the requested revision of how do these materials behave under pure SO₂?

I have looked up some related MOF-SO₂ chemistry and can see that NOTT-300 has an uptake of some 8 mmol/g (under pure SO₂) at 1 bar and although the current system has a higher uptake than NOTT-202 (as cited in the manuscript) no mention is made of the higher capacity (and stability?) of NOTT-300, although comparisons of isosteric heats of adsorption are made. These will be higher in the current doped systems, I assume due to a degree of chemi- rather than physic-sorption taking place.

I also note that there are no PXRD patterns for SO₂ loaded MOF in the manuscript nor in SI. This is again surprising given the reported stability of the material; showing patterns pre and post adsorption is not quite the same.

I have also read the other referee's comments with which I entirely concur.

I have read this manuscript many times now and I am well aware how frustrating it can be for authors when referees "pick holes" at results. I am trying very hard not to do that here, but the story being told here (to me) is not entirely clear. The analogous chemistry with 1-3, 1-KOH to 3-KOH with CO₂ has been reported previously, and the current paper now reports 1-Ba(OH)₂ to 3-Ba(OH)₂ with SO₂. It is a nice piece of work but on balance I do not think it quite makes it for Nature Comm in terms of genuine novelty and full characterisation as discussed above. If I am out of line with other referees then I am very content to be overruled.

Reviewer #3 (Remarks to the Author)

The authors have now sufficiently addressed my prior issues. I believe the contribution is sufficiently novel and likely to be of interest to a wide cross section of the MOF community.

Reviewer # 2 comments:

I requested that isotherms for SO₂ uptake over multiple cycles be included. This required extending the experiment already shown in Figure S8 (which has a strange shape actually) to show multiple sorption - cycles desorption so that it is absolutely clear and unambiguous what the level of reversibility is in this system. The authors have chosen not to do this but show breakthrough cycles instead. This still raises the confusion in my mind as to the levels of chemi- and physi-sorbed SO₂ in these materials and what is the behaviour of the material in terms of formation of BaSO₃ inside or outside the MOF pores. The new experiments are conducted on N₂/SO₂(97.5:2.5) flow (ie highly diluted SO₂ streams) and so does not cover the requested revision of how do these materials behave under pure SO₂?

Answer:

As indicated above, we have now measured XRPD patterns and FTIR spectra of SO₂ loaded samples at 1 bar of SO₂. The results agree with SO₂ adsorption taking place in MOF particles only, since Ba(OH)₂ XRPD pattern remain unaltered, while, FTIR is in agreement with coexistence of physisorbed (major component) and chemisorbed SO₂. These results are also in agreement with materials stability towards SO₂ adsorption (see p. 5 first paragraph and supplementary Fig. 10, 12).

Besides, there is no evidence of BaSO₃ formation in the cocrystallized Ba(OH)₂ which agrees with a clearly differentiated reactivity of Ba extraframework cations in the MOF structure compared to unreactive cocrystallized Ba(OH)₂ particles (p. 5 first paragraph and Supplementary Fig. 12).

Regarding the comment of why are we more interested in carrying out the SO₂ capture measurements with low relative pressure of this gas instead under pure SO₂, the main reason is because the principal objective of the present work is to simulate real application conditions in which SO₂ will be highly diluted in a complex composition flue gas. In this regard, it should be noted that adsorption capacity, although important, it is not the most important parameter in a gas separation process but adsorption selectivity. Indeed, the strongest point of our postsynthetically modified materials is in terms of adsorption selectivity, namely SO₂/CO₂ partition coefficients are 1 order of magnitude higher than NOTT/MTM-300 materials (see p.8 first paragraph) and give rise to efficient capture of SO₂ even in the presence of highly competitive adsorbates such as CO₂ and H₂O which is highly relevant in a real separation process (p. 6 last paragraph).

Regarding the XRPD patterns of SO₂ loaded materials. As indicated above we have measured the XRPD pattern of our material under pure SO₂ with the results being indicative that the material remains unaltered during and after SO₂ exposition. This result is not surprising taking into account the rigidity and stability of the twelve connected framework.

Reviewer #2 comment:

I have read this manuscript many times now and I am well aware how frustrating it can be for authors when referees "pick holes" at results. I am trying very hard not to do that here, but the story being told here (to me) is not entirely clear. The analogous chemistry with 1-3, 1-KOH to 3-KOH with CO₂ has been reported previously, and the current paper now reports 1-Ba(OH)₂ to 3-Ba(OH)₂ with SO₂. It is a nice piece of work but on balance I do not think it quite makes it for Nature Comm in terms of genuine novelty and full characterisation as discussed above. If I am out of line with other referees then I am very content to be overruled.

Answer: Our previous report on the chemistry and adsorptive properties of 1-3, 1@KOH to 3@KOH materials was limited to CO₂ capture. In this contribution, we report three additional new materials 1@Ba(OH)₂, 2@Ba(OH)₂, 3@Ba(OH)₂ which have been obtained employing an specific ion exchange strategy with barium salts aimed to the selective capture of SO₂. Moreover, the SO₂ adsorption/separation properties have been studied along the whole 1-3, 1@KOH-3@KOH, 1@Ba(OH)₂-3@Ba(OH)₂ series, which has been a challenging work taking into account the corrosive and toxic nature of this gas. In addition, the density functional computational study showing both the location and structure of crystal defects, along with the interaction of adsorbate molecules with crystal defect sites, is a completely new work and a very challenging one.

Summarizing, we are happy with the positive input of the reviewers comments that have certainly contributed to improve the quality of our contribution.